

# Hand grab or noose pole? Evaluating the least stressful practice for capture of endangered Turks and Caicos Rock Iguanas *Cyclura carinata*

Giuliano Colosimo[1], Gwyneth Montemuro[2], Gregory A. Lewbart[3], Gabriele Gentile[1] and Glenn Gerber[4]

[1] Department of Biology, University of Rome Tor Vergata, Rome, Lazio, Italy
[2] School of Veterinary Medicine, St. Mathew University of Grand Cayman, West Bay, Cayman Island
[3] College of Veterinary Medicine, North Carolina State University, Raleigh, North Carolina, United States
[4] San Diego Zoo Wildlife Alliance, Escondido, California, United States

Corresponding author
Giuliano Colosimo,
giuliano.colosimo@uniroma2.it

## ABSTRACT

As the analysis of blood metabolites has become more readily accessible thanks to the use of point-of-care analyzers, it is now possible to evaluate stress level of wild animals directly in the field. Lactate is receiving much attention as a good stress level proxy in individuals subjected to capture, manual restraint, and data sampling in the wild, and appropriate protocols to maintain lactate values low should be preferred. In this study we compare how two different capture methodologies, hand grab *vs.* noose pole, affect the variation of blood lactate values in *Cyclura carinata* iguanas when captured for sampling. We used blood lactate concentration, measured immediately upon- and 15 min after-capture, as a proxy for stress level. While the primary goal of this work is to determine the least stressful capture methodology to be favored when sampling this and other wild iguanas, we also evaluated additional baseline physiological parameters relevant to the health and disease monitoring for this species. Our results show that while initial lactate values level-out in sampled individuals after 15 min in captivity, regardless of the capture methodology, rock iguanas captured by noose pole showed significantly higher lactate concentration and increased heartbeat rate immediately after capture. While the overall health evaluation determined that all analyzed individuals were in good health, based on our results we recommend that, when possible, hand capture should be preferred over noose pole when sampling wild individuals.

## INTRODUCTION

Point-of-care analyzers are becoming the new standard for the quick and precise assessment of important physiological parameters in human as well as veterinary medicine (*Lubba et al., 2011*; *Park, 2021*; *Stern & Camus, 2022*). For field biologists, ecologists, and

conservationists this is a great leap forward as these devices allow monitoring of blood metabolites directly in the field and making a first assessment of the overall clinical and health status of wild individuals and populations at risk of extinction. Veterinary health examinations in species of conservation concern can dramatically improve the survival of individuals and populations, as they can be used to quickly detect health related threats and trigger the appropriate conservation response (*Walton, 2001*; *Colosimo et al., 2022*; *Lewbart et al., 2019*).

Health assessments entail checking vital signs, recording morphometric data, and collecting fresh blood samples for analysis in the field and laboratory. For health examinations to remain both ethical and beneficial it is also important to consider how the method of capture and restraint of sampled individuals can affect their overall welfare. To this end, among the many parameters measured by point-of-care analyzers, lactate is gaining much attention as it is often used as a proxy to evaluate stress level, generally considered an indicator of animal welfare, of sampled individuals (*Trondrund et al., 2022*). A number of studies have shown how lactate levels in the blood can be used as a diagnostic tool for chronic conditions, but also as an indicator of stress associated with struggling, fear, and/or bursts of intense muscular activity in a number of species (*Rand et al., 2002*; *Klein et al., 2020*; *Trondrund et al., 2022*). Lactate is a metabolite generally produced from pyruvate as a byproduct of glycolysis. In aerobic conditions pyruvate, normally generated during glucose metabolism, is processed and decarboxylated in mitochondria (*Pang & Boysen, 2007*). In anaerobiosis, pyruvate can no longer undergo decarboxylation and lactate is produced as a byproduct. While lactate is produced by most body tissues, skeletal muscle is heavily involved in its yield (*Pang & Boysen, 2007*). As a result, short and intense bursts of activity can dramatically increase the lactate concentration in the blood stream and other tissues (*Pang & Boysen, 2007*). A diagnostic approach that uses blood lactate concentration level to infer stress level while sampling wild and captive individuals has been developed and used in mammals, but numerous studies have now shown its utility in reptiles as well (*Klein et al., 2020*; *Mones et al., 2021*; *Molinaro et al., 2022*).

Ectothermic vertebrates, such as reptiles, generally have a small aerobic scope and obtain a large proportion of their energy associated with routine daily activities through anaerobic metabolism (*Gleeson, 1991*). In the common green iguana, *Iguana iguana*, at least three-fourths of the energy used during daytime routine is derived from anaerobiosis and, hence, produces lactic acid (*Moberly, 1968*). The baseline lactate concentration of reptiles is known to rapidly rise after intense physical activity. In juvenile Galapagos marine iguanas, *Amblyrynchus cristatus*, lactate concentration rose approximately 32% after 2 min of forced and sustained swimming activity (*Bennet, Dawson & Bartholomew, 1975*; *Bartholomew, Bennet & Dawson, 1976*. A similar pattern of lactate accumulation has been observed in the desert iguana, *Dipsosaurus dorsalis*, after 15 s of running at 1 m/s (*Donovan & Gleeson, 2006*). A short and intense burst of activity is what is experienced by many reptile species during routine and Institutional Animal Care and Use Committee (IACUC) approved procedures for field sampling and restraining (*Goessling & Mendonça, 2020*; *Klein et al., 2020*; *Molinaro et al., 2022*). For example, *Mones et al. (2021)* conducted a study on loggerhead sea turtles (*Caretta caretta*) comparing the performance of two point-

of-care analyzers after manually restraining individuals. The authors suggest that blood gas and lactate levels should be taken promptly after capture to acquire the most accurate measurement and suggest limited handling to minimize overall impact of health examinations on animals (*Mones et al., 2021*). Clinically, concern pertains to susceptibility for lactate acidosis in reptiles that are stressed, causing their lactate threshold to be surpassed. Currently, there is little available data on what could constitutes a lactate threshold in most reptile species. However, despite their unique compensatory blood barrier system, concealing their state of lactate acidosis, reptiles with higher concentrations of lactate are physiologically exhausted and unable to respond to stimuli, and this condition could trigger other complications, such as respiratory diseases, making mindful and scrutinous health care vital for this group of organisms (*Prezant & Jarchow, 1997*; *Divers & Stahl, 2019*).

In the present study we compare how two alternative capture techniques (hand grab *VS* noose pole), generally adopted to sample *Cyclura* rock iguanas, affect the overall lactate accumulation in sampled individuals. We took advantage of a long term and ongoing monitoring program on *Cyclura carinata* iguanas in the Turks and Caicos Islands (TCI) and analyzed how these capture methodologies influence blood lactate levels of rock iguanas over a 15 minute time interval. In doing so we wanted to assess the capture/handling impact and determine the least detrimental capturing technique. We hypothesized that the hand grab methodology would constitute, overall, a less-stressful approach as sampled individuals struggle less when they are captured directly by hand (see Materials and Methods). This study was performed within the framework of a general health screening of wild Turks and Caicos Rock Iguanas. While also reporting on the results of these health examinations we discuss our findings and suggest best sampling practices for this and other species of iguanas.

# MATERIALS AND METHODS

## Ethic statement

This study was conducted on several islands in the Turks and Caicos Islands with permission of the TCI Department of Environment and Coastal Resources (Scientific Research Permits 2021-06-10-09 and 2022-07-04-29) and approval of the San Diego Zoo Wildlife Alliance IACUC (Proposal #21-011). In conducting the study, the authors abided by the national "Animal health ordinance" (Revised edition 31 March 2018). The nature of this study implied that only two of the 3R tenet of animal welfare (Replacement, Reduction, and Refinement; *Field et al., 2019*) could be adopted. We could not avoid or replace the target animal species (Tenet 1, *Field et al., 2019*), but we did limit our sampling to the minimum number of individuals that would let us reach a meaningful and statistically sound result (Tenet 2, *Field et al., 2019*). Finally, the sampling procedure was carried out by a licensed Doctor of Veterinary Medicine and co-author of this study (GAL), and each specimen was immediately released, with no harm, at the end of the sampling procedure.

## Study system

*Cyclura carinata* are endemic to the TCI. This species inhabits ca. 75 of the over 250 islands and cays in the TCI, and recent genetic analyses identify two separate Evolutionary Significant Units (ESUs) and multiple Management Units (MUs) across their range (*Welch et al., 2017*). Despite a recent status downgrade from critically endangered (CR; *Gerber, 2004*) to endangered (EN; *Gerber, Colosimo & Grant, 2020*) in the International Union for Conservation of Nature (IUCN) Red List of Threatened Species, the species remains extirpated from 90% of its historic range and *C. carinata* is still in dire need of conservation actions (*Havery et al., 2021*). The survival and persistence of these iguanas are directly threatened by invasive feral mammals (cats and dogs in particular; *Iverson, 1978*; *Gerber, Colosimo & Grant, 2020*; *Gerber, 2004*), and by the harmful impact of human activities and development (*Gerber, 2007*; *Havery et al., 2021*). *Cyclura carinata* are the largest native terrestrial vertebrates and dominant herbivores in the TCI. They act as seed dispersers for a variety of endemic plants (*Auffenberg, 1982*; *Iverson, 1985*; *Olesen & Valido, 2003*) and serve as a flagship species in the fight against biodiversity loss and promotion of habitat protection (*Knapp, 2005*). Unfortunately, habitat degradation and loss are accelerating due to rapid human expansion (*Havery et al., 2021*). Greater enforcement of laws, better regulation of development, improved biosecurity, and expanded control of invasive species would benefit Turks and Caicos rock iguanas and their habitat. Alongside programs to improve biosecurity, control invasive species, and reduce habitat destruction, veterinary health examinations of this keystone species can improve the survival of individuals and populations.

## Data sampling

Our study included TCI iguanas captured during two field expeditions in February and August 2022, respectively. During the first expedition we sampled two iguanas from Pine Cay, 21 from Little Water Cay, three from Lizard Cay, and four from Bird Cay. Fifteen iguanas from Big Ambergris Cay were sampled during the second expedition (Fig. 1). Priority was given to adult individuals, but, as this study falls within a long-term species monitoring program, we also collected samples from juveniles (see below).

Two different methods of capture were adopted: hand grab and noose pole. These two methodologies are usually the most commonly used, even though the use of traps and nets are sometimes adopted. The first method entails luring individuals with pieces of vegetables or fruit dropped at the feet of the researcher. As the iguana approaches and is distracted attempting to eat the bait, it is captured by the researcher using gloved hands. With this protocol the sampled individual has little to no struggling time, as it is pinned down during the hand-grabbing procedure. The second capturing technique involves the use of a noosed telescopic pole. This approach allows the capture of iguanas up to 4 m away, and works extremely well with näive iguanas (*i.e.*, those individuals that have never been captured before). When the iguana realizes it has been captured in the noose it starts to struggle, trying to run away and sometimes performing what is known as the "alligator-roll". There usually is a 1–5 s lag time before the researcher reaches the individual and
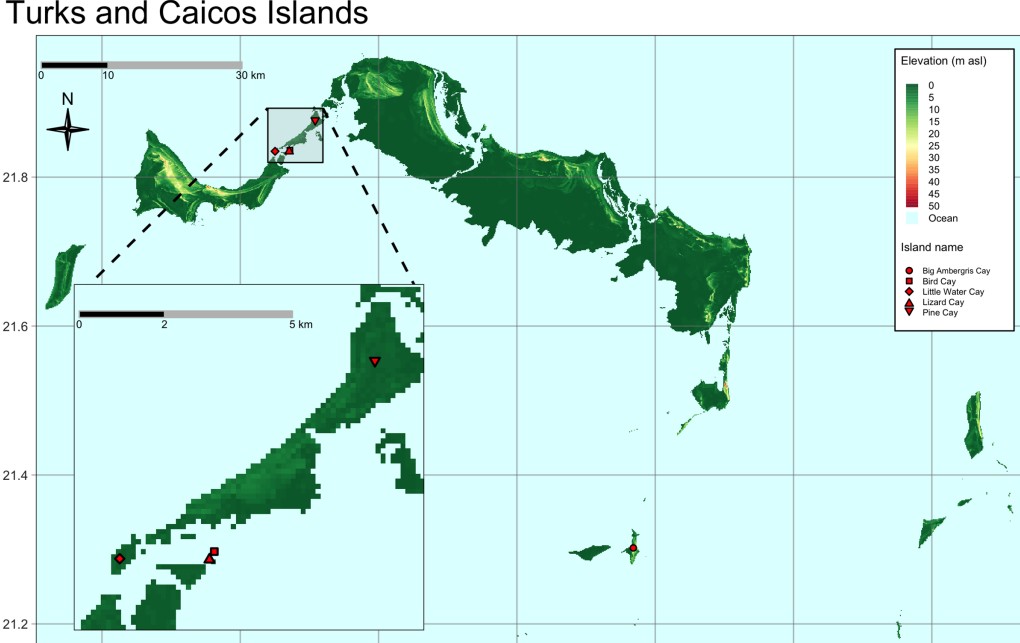

**Figure 1 Map of the Turks and Caicos Islands, located north of Hispaniola at the southeastern terminus of the Bahaman Archipelago.** Sampled cays and populations are highlighted with red symbols. Map inset shows in greater detail the location of the cays sampled on the east side of the Caicos Bank. Bird Cay and Lizard Cay are very small in size and close to each other.

grabs it using gloved hands, pinning it down as in the previous procedure and removing the noose.

Immediately following capture, each iguana was manually restrained, and the first of two blood samples was collected by venipuncture of the coccygeal vein at the hemal arch of the tail (within 3 min of capture). A 22-gauge 3.8 cm needle with a 3 mL heparinized syringe was used to extract between 1–2 mL of blood. The blood was immediately divided into sub-samples. A couple drops were used for making blood films on clean glass microscope slides. About 0.1 mL was used to estimate lactate concentration using a portable Lactate Plus™ analyzer (Nova Biomedical, Waltham, MA, USA), and another 0.1 mL was loaded into an iSTAT Portable Clinical Analyzer (Heska Corporation, Fort Collins, Colorado, USA) using CHEM8$^+$ cartridges to estimate total carbon dioxide ($tCO_2$), hematocrit (Hct; hematocrit values were also recorded using an analogous procedure involving a microcentrifuge, HctM), hemoglobin (Hb), sodium (Na), potassium (K), chloride (Cl), ionized calcium (iCa), creatinine (Crea), blood urea nitrogen (BUN), anion gap (AG), and glucose (Glu). Finally, about 0.05 mL was used for centrifugation with a portable microcentrifuge to determine hematocrit and total solids. The latter value was determined with a refractometer utilizing a drop or two of plasma from the hematocrit tube.

After collecting the first blood sample, health examinations were performed. Core body temperature was measured using an ERBO compact J/K/T/E with a T-PVC epoxy-tipped

24 GA probe at the cloaca. A Doppler ultrasound probe (Parks Medical Electronics, Inc., Aloha, OR, USA) was pressed over the heart to record heart rate. Snout-vent length (SVL) and tail length (TL) were obtained by stretching the ventral midline of each iguana along a semi-rigid, clear plastic ruler. Body mass was obtained using the smallest suitable Pesola spring scale (Pesola Präzisionswaagen AG, Schindellegi, Switzerland). Individuals captured for the first time received a passive integrated transponder (PIT) tag sub-dermally in the left rear leg for permanent identification, and a unique, colored bead tag in the nuchal crest for temporary visual identification (*Rodda et al., 1988*). After 15 min in captivity, whether the health examination was concluded or not, a second 0.2–0.5 mL blood sample was drawn to determine the second lactate value. The blood remainder was stored in blood-buffer for future genetic analyses. Following health examinations and processing, all iguanas were released at their original point of capture.

## Data analysis

Data were analyzed using R version 4.3.1 (*R Core Team, 2023*) running within RStudio version 2023.09.0 + 463 (*R Studio Team, 2023*).

## Stress response to capture method

We investigated the stress response to different sampling techniques focusing on lactate concentration and heart rate data. We organized our lactate data in a two (hand grab *VS* noose pole) by two ($t_0$ and $t_{15}$) table design, with the first factor (capture methodology) varying between individuals (*i.e.*, some individuals were captured by hand and some by noose), and the second factor (sampling time) varying within individuals (*i.e.*, in each individual lactate was measured at two different time points, and the collected data inherently includes individual variation in lactate production). We first measured descriptive statistics (sample mean ± 1 standard deviation) in each one of these four groups while also accounting for sex. We looked for significant differences between males and females captured with alternative techniques and at different time points using a Wilcoxon Rank Sum Test. Lactate measurements were then used as the dependent variable in a linear mixed effect model using capture methodology and time as independent variables. The ID of individual iguanas was used as a random effect to account for the non-independence of error distribution in our data. This allowed us to produce results that controlled for within individual differences in lactate production. To also account for potential differences in lactate production due to body condition (see below) and sex we also included these two variables as fixed effect covariates. We used the *lmer4* R package (*Bates et al., 2015*) to fit the model using a restricted likelihood method and we tested for model assumptions using the *performance* R package (*Lüdecke et al., 2021*). When running the model as previously described we found residuals to be heteroscedastic (*i.e.*, unequal error variance with $p$-value = 0.026). We first tried to log transform the response variable and re-run the model. While this approach resolved the heteroscedasticity problem it made it so that the variance explained by the random effect variable was estimated at- or very close to 0, confounding the interpretation of the results. Therefore, we proceeded with the computation of two additional reduced models without log transforming the response

variable and removing sex or condition index, respectively, in each model. In both cases, model residuals were homoscedastic and presented no other issues as demonstrated by the results of applying the *allFit()* function from *lmer4* R package (*Bates et al., 2015*) to assess that all linear mixed model requirements were satisfied in the result. We computed the Akaike Information Criterion (AIC) to retain the model that would best describe the data.

Heart rate can be considered another proxy of stress level. As heart rate was only recorded once, after the collection of the first blood sample, we built a simple linear regression model using log transformed heart rate values as the response variable and capture methodology, body temperature and body condition index as covariates. We further computed estimated marginal means (EMMs), using the *emmeans* R package (*Lenth, 2023*), to specifically evaluate the effect of capture methodology on stress response as measured by heart beat rate.

## Health examination

To describe the physical condition of individual iguanas we used morphological measurements to calculate a body condition index (BCI) following *Peig & Green (2009)*. The usefulness of BCIs heavily depends on the definition of condition index adopted (*Labocha, Schutz & Hayes, 2014*). The BCI suggested by *Peig & Green (2009)*, called scaled mass index (SMI), can be interpreted as the nutritional state of an individual (*i.e.*, the energy an individual accumulated through foraging). The SMI is able to account both for structural differences in the size of individuals (for example, sexual dimorphism between males and females, see below for calculation details), while also considering the size of the energy stores, expressed by simple mass measurements, in sampled individuals. We used the formula $SMI = M_i[L_o/L_i]^{b_{sma}}$, where $M_i$ and $L_i$ are mass and SVL, respectively, of individual $i$; the scaling exponent $b_{sma}$ is estimated using a standardized major axis (SMA) regression of mass on SVL; $L_0$ is an arbitrary value for SVL (in this particular case we used the average SVL value estimated from sampled individuals). The SMI value can be interpreted as the predicted mass measurement of an individual when its linear body value is standardized to $L_0$ (*Peig & Green, 2009*). We accounted for species sexual dimorphism (*Iverson, 1977*) by estimating two scaling exponents, one for each sex, using the *smatr* R package (*Warton et al., 2012*).

We calculated descriptive statistics (sample mean ± 1 standard deviation, minimum and maximum values), in males and females separately, for all blood parameters measured. The data obtained were compared to those available in other iguanid species. We assessed significant differences between sexes using nonparametric Wilcoxon Rank Sum tests and we explored the correlation between every measured variable adjusting for multiple comparisons using false discovery rate (FDR) (*Benjamini & Hochberg, 1995*).

## RESULTS

All 45 iguanas were considered to be in satisfactory health by a board-certified zoological medicine veterinarian and co-author in this study (GAL). During physical examinations, subjects remained relatively calm. Of the 45 iguanas captured for this study, lactate samples were successfully collected from 42 (20 females and 22 males). Of these, 19 (eight

females and 11 males) were captured by hand and 23 (12 females and 11 males) were captured using a noose pole. Females captured by hand had a mean initial lactate value ($t_0$) of 5.15 (±1.79) mmol/L, while females captured by noose pole had a mean initial lactate value of 9.09 (±3.56) mmol/L. A similar pattern was observed for male iguanas. Initial lactate levels for males captured by hand had a mean of 6.69 (±3.56) mmol/L, while males captured using a noose pole had a mean of 11.30 (±3.45) mmol/L. In both females and males average lactate concentration between individuals captured *via* hand grab or noose pole were statistically significant using a Wilcoxon Rank Sum Test ($p$-val = 0.005 for females and $p$-val = 0.02 for males). For the second blood draw ($t_{15}$) average lactate values were similar across all groups of samples and no statistical differences were reported. Females caught by hand presented, at $t_{15}$, a mean of 16.00 (±3.22) mmol/L, while females caught by noose had a mean of 15.8 (±2.51). Males caught by hand averaged 15.20 (±2.32) mmol/L and those caught by noose averaged 15.90 (±4.14) mmol/L (see Fig. 2 for a graphical representation of the data and their distribution). In one of the 42 successfully sampled individuals the second lactate reading was not available due to a lactate strip malfunction during the data recording procedure.

Of the two reduced models computed, the one with the lowest AIC included capture methodology, sampling time and SMI as fixed effect covariates. The selected model indicated a significant effect of all variables (model results are available in Table 1). Capture methodology also significantly affected HR. Individuals sampled by hand had an average HR of 84 beats per minute (bpm) (±19.5) while those sampled by noose averaged 101 bpm (±19.3). The linear model indicated a strong and significant effect of capture methodology on HR differences (t-value = 2.309, $p$-val = 0.027), and body temperature strongly influenced HR as well (t-value = 4.530, $p$-val << 0.01). Neither sex nor body condition, as estimated by the SMI, influenced HR. The marginal contrast analysis built using estimated marginal means of heart rate controlling for the other covariates of the linear model revealed an average difference of 11 heart beats per minute (95% CI [0.91–22.52]) between individuals captured by hand and those captured using a noose. This difference was significant ($p$-val = 0.034).

Descriptive statistics for morphological features and biochemical parameters are summarized in Tables 2 and 3 and graphically represented in Fig. 3. Analysis of SMI body condition index confirmed the results from the overall physical examination (*i.e.*, all animals were in satisfactory health and physical condition) with the exception of one outlier individual, that was in suboptimal health and physical condition (Fig. S1). No significant differences were found between any of the biochemical parameters measured from blood samples between males and females in any of the comparisons performed. After adjusting significance of correlation tests using the FDR approach (*Benjamini & Hochberg, 1995*) we found numerous parameters to be significantly correlated (Fig. 4).

## DISCUSSION

The conservation challenges that many species are facing worldwide is fostering the need to capture animals for management and research purposes. In order to promote animal
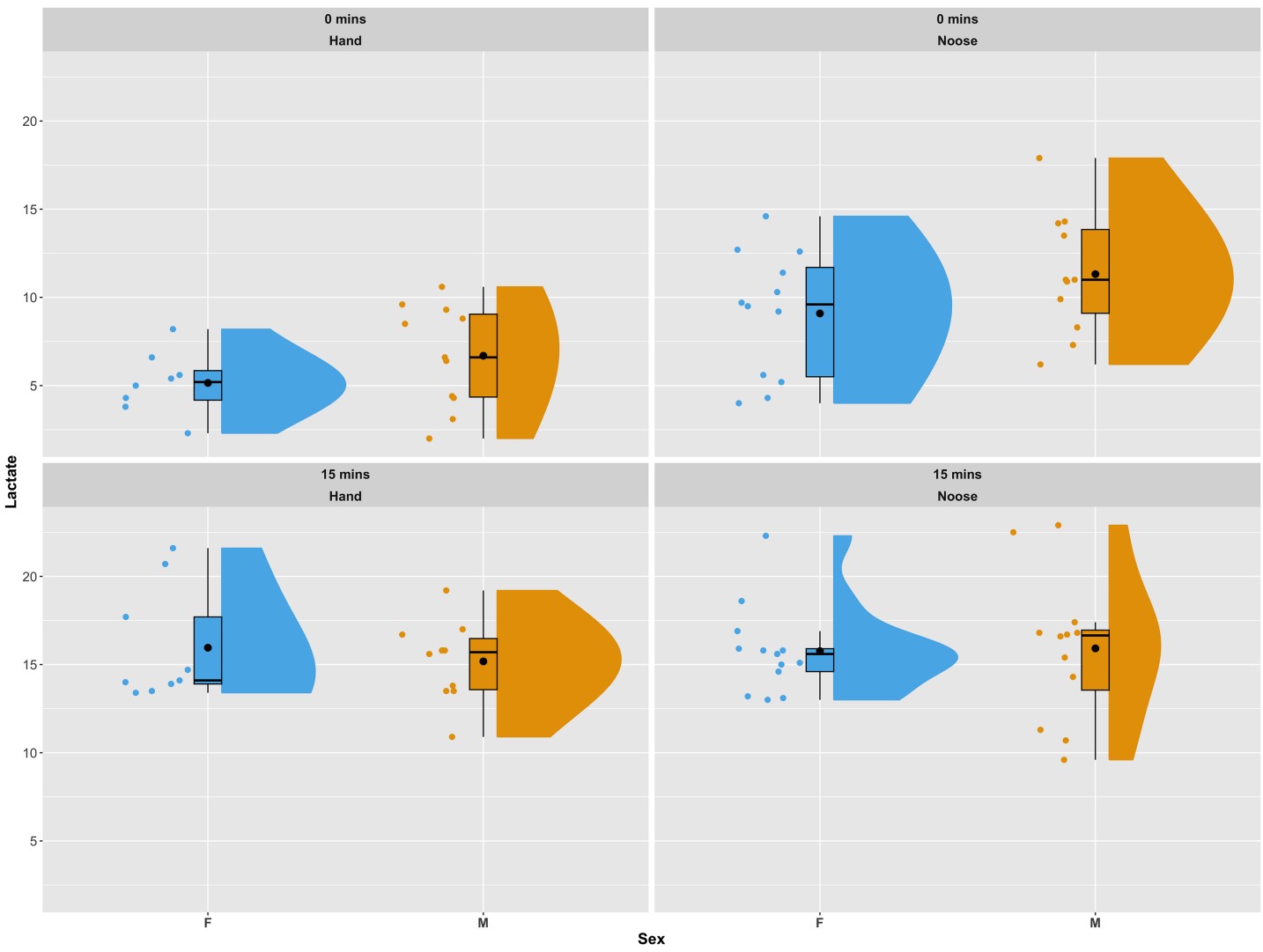

**Figure 2 Distribution of lactate values in sampled individuals as a combination of a boxplot, a kernel density plot and a scatter plot.** The boxplots show the median (horizontal black line) and the mean value (a black dot), the interquartile range (colored box), and the range (vertical black line). The kernel density plots provide a non-parametric estimate of the data distribution. The top panels present lactate values for the first blood draw ($t_0$). The bottom panels show lactate values for the second blood draw ($t_{15}$). Panels on the left side show individuals captured by hand, while panels to the right show individuals captured using a noose pole. Data are displayed and colored according to sex, with blue for females and orange for males. The plots were generated using the *plot_raincloud()* R function from the sdamnr package (*Speekenbrink, 2022*).

welfare, it is important to assess how capture techniques may influence the overall physiological status of sampled individuals. In this study we compared two capture techniques regularly used to catch Turks and Caicos rock iguanas (hand grab and noose pole) and used lactate concentration in blood collected from sampled individuals to determine which capture technique has the least stressful impact on these iguanas. Our results show that male and female iguanas captured without resorting to a noose pole present significantly lower initial ($t_0$) blood lactate concentration values (5.15 *VS* 9.09 mmol/L for females; 6.69 *VS* 11.30 mmol/L for males). Interestingly, our results

**Table 1 Linear mixed effect model with lactate concentration as the response variable.** The first column shows the independent and fixed terms: capture method with noose used as reference (CM_NOOSE), time of blood sampling with $t_{15}$ used as reference (TIME_T15), and body condition index (SMI). Other columns show the results of the model: parameter estimate or slope of the correlation (Estimate), standard error (St. Err.), degrees of freedom (df), T-statistic (t), significance of the estimated variation (*p*-val). The model includes a random term (individuals' ID) with average variance of 2.193 and standard deviation of 1.481.

| | Estimate | St. Err. | df | t | *p*-val |
|---|---|---|---|---|---|
| CM_NOOSE | 2.257 | 0.751 | 39.313 | 3.003 | 0.004 |
| TIME_T15 | 7.352 | 0.601 | 40.326 | 12.238 | << 0.01 |
| SMI | 0.003 | 0.001 | 39.078 | 2.577 | 0.013 |

**Table 2 Morphometric and physiologic parameters for male and female Turks and Caicos rock iguanas (*Cyclura carinata*) obtained during routine health examinations.** Values are shown for number of individuals sampled (n), parameter means (±standard deviation), and range of parameter values. Snout vent length (SVL, in mm), mass (in g), heart rate (HR, in beats per minutes), temperature (T, in °C) and scaled mass index (SMI).

| | Females | | | Males | | |
|---|---|---|---|---|---|---|
| Measure | *n* | Mean (±sd) | Range | *n* | Mean (±sd) | Range |
| SVL | 22 | 233.63 (44.32) | 106.00–313.00 | 23 | 292.60 (71.81) | 99.00–390.00 |
| Mass | 22 | 585.77 (297.23) | 51.00–1,510.00 | 23 | 1,281.80 (811.97) | 35.50–2,845.00 |
| HR | 22 | 91.63 (19.43) | 66.00–132.00 | 23 | 95.47 (22.73) | 60.00–138.00 |
| T | 19 | 34.34 (4.80) | 26.70–44.90 | 22 | 34.48 (4.30) | 26.20–40.90 |
| SMI | 22 | 538.00 (84.40) | 296.00–659.00 | 23 | 1,075.00 (132.00) | 849.00–1,341.00 |

**Table 3 Blood biochemistry parameters for male and female Turks and Caicos rock iguanas (*Cyclura carinata*) obtained during routine veterinary health examinations.** Values are shown for number of individuals sampled (n), analyte means (± standard deviation), and range of values determined using an i-STAT analyzer. Metabolite values shown for sodium (Na), potassium (K), chloride (Cl), ionized calcium (iCa), total carbon dioxide ($tCO_2$), glucose (Glu), hematocrit (Hct), hematocrit measured with a microcentrifuge (Hct-M), hemoglobin (Hb), anion gap (AG), and total solids (TS).

| | Females | | | Males | | |
|---|---|---|---|---|---|---|
| Metabolite | *n* | Mean (±sd) | Range | *n* | Mean (±sd) | Range |
| Na (mmol/L) | 22 | 156.86 (4.74) | 150.00–168.00 | 22 | 160.68 (5.84) | 144.00–171.00 |
| K (mmol/L) | 21 | 3.73 (1.28) | 2.00–7.30 | 16 | 3.78 (0.83) | 2.30–5.10 |
| Cl (mmol/L) | 22 | 124.90 (5.68) | 113.00–134.00 | 22 | 124.90 (5.87) | 111.00–133.00 |
| iCa (mmol/L) | 22 | 1.37 (0.14) | 1.10–1.73 | 22 | 1.38 (0.20) | 0.77–1.74 |
| $tCO_2$ (mmHg) | 22 | 17.40 (5.59) | 9.00–31.00 | 22 | 18.09 (5.85) | 10.00–30.00 |
| Glu (mg/dL) | 22 | 150.27 (22.92) | 103.00–206.00 | 22 | 150.00 (25.64) | 120.00–213.00 |
| Hct (%) | 22 | 23.63 (3.79) | 18.00–30.00 | 22 | 26.09 (5.07) | 18.00–40.00 |
| Hct-M (%) | 20 | 29.55 (5.52) | 22.00–45.00 | 22 | 30.11 (4.96) | 22.00–40.50 |
| Hb (g/L) | 22 | 8.03 (1.29) | 6.10–10.20 | 22 | 8.87 (1.72) | 6.10–13.60 |
| AG (mEq/L) | 22 | 19.14 (5.17) | 13.00–32.00 | 16 | 20.75 (5.70) | 12.00–35.00 |
| TS (g/L) | 20 | 8.24 (1.40) | 6.00–10.80 | 18 | 8.36 (1.37) | 6.50–10.90 |

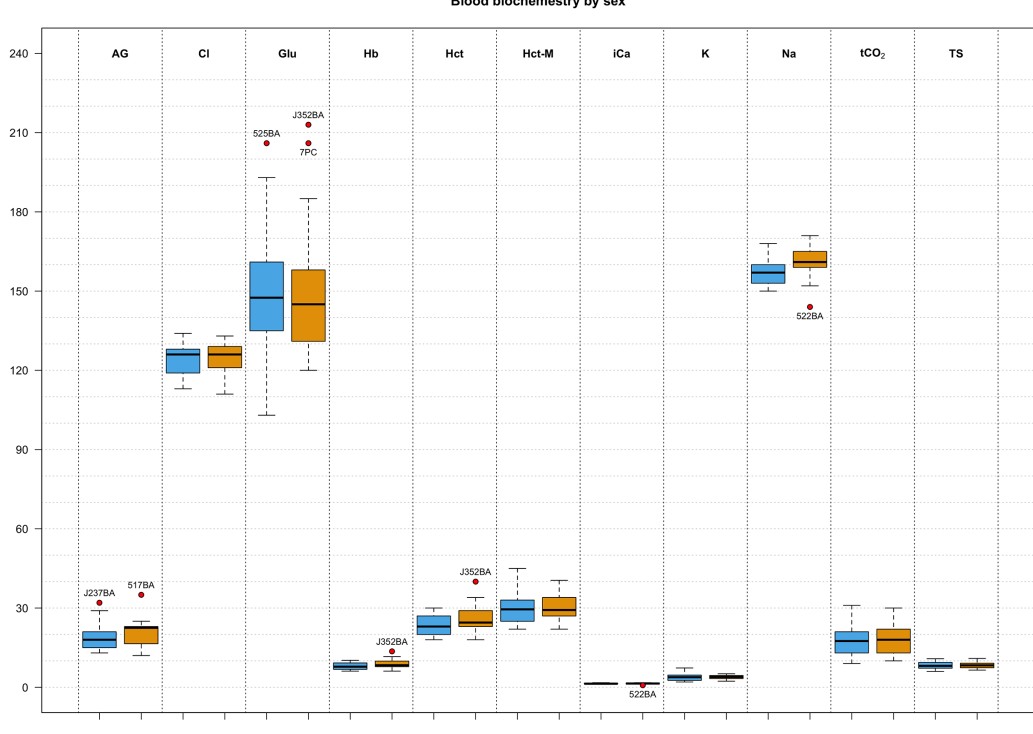

**Figure 3 Box and whisker plots of biochemistry parameters measured for Turks and Caicos rock iguanas.** The boxplots show the median (horizontal black line), the interquartile range (colored box), and the range (vertical black line). Red dots outside the range represent outlier values and their unique animal identifier. Values for females are highlighted in blue while values for males are highlighted in orange. The biochemical parameters in this figure are: anion gap (AG, in mEq/L), chloride (Cl, in mmol/L), glucose (Glu, in mg/dL), hemoglobin (Hb, in g/L), hematocrit (Hct, in %), hematocrit measured with a microcentrifuge (Hct-M, in %), ionized calcium (iCa, in mmol/L), potassium (K, in mmol/L), sodium (Na, in mmol/L), total carbon dioxide (tCO$_2$, in mmHg), and total solids (TS, in g/L).

indicate that using a noose pole also significantly increases the HR frequency in captured animals (88.1 *VS* 100.1 mean bpm for females; 81.5 *VS* 92.6 mean bpm for males).

Overall, the amount of stress exerted on *C. carinata* iguanas during capture procedures can be considered low and while increased lactate levels at t$_0$ for animals captured using a noose pole can be attributed to stress from capture there were no apparent underlying contributing health concerns. Currently, lactate values at rest or during routine activities are available only for a handful of iguanid species. In the desert iguana, *Dipsosaurus dorsalis*, several studies have analyzed recorded blood lactate concentration under different levels of physical activity. For example, *Donovan & Gleeson (2006)* found an average concentration of blood lactate of 1.1 mmol/L in individuals kept in controlled conditions prior to an experimental treatment (*Donovan & Gleeson, 2006*). In a previous study, *Scholnik & Gleeson (2000)* found blood lactate values at rest averaged 3.5 (± 1.4) mmol/L for individuals of the same species. Blood lactate concentration rose to above 35 mmol/L only after a short but intense burst of physical activity, after which individuals needed
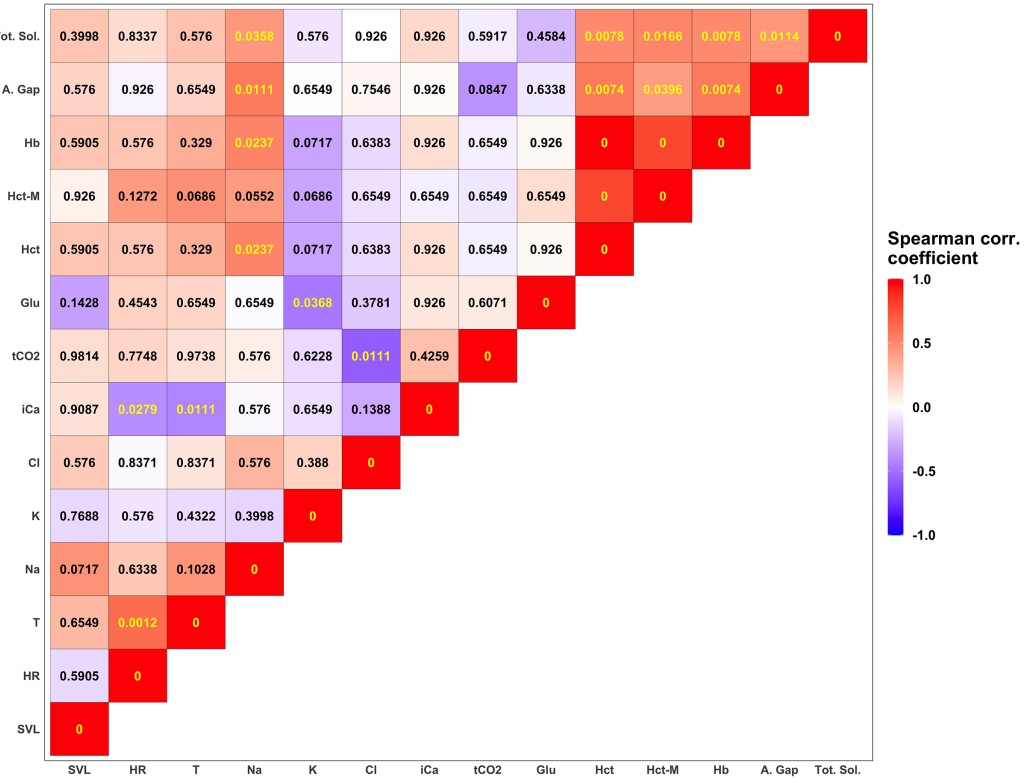

**Figure 4  Summary of correlation between measured parameters.** Each tile is colored according to the strength of the correlation between measured variables using the Spearman correlation coefficient. Darker shades of blue correspond to increasingly negative correlations while warmer shades of red correspond to increasingly positive correlations. Numerical values in each tile correspond to the p-value corrected using the FDR approach (*Benjamini & Hochberg, 1995*). *P*-values lower than the significant threshold of 0.05 are colored in yellow, while non-significant correlations are colored in black. The biochemical parameters in this figure are anion gap (A. Gap), chloride (Cl), glucose (Glu), hemoglobin (Hb), hematocrit (Hct), hematocrit measured with a microcentrifuge (Hct-M), ionized calcium (iCa), potassium (K), sodium (Na), total carbon dioxide ($tCO_2$), and total solids (Tot. Sol.).

almost 2 h to recuperate and re-establish baseline values (*Scholnik & Gleeson, 2000*). In our study the maximum lactate value, recorded at $t_{15}$, was 22.9 mmol/L.

Other studies in reptiles have shown that lactate concentration values are generally positively correlated with time since capture or since the onset of an activity burst (*Bartholomew, Bennet & Dawson, 1976*; *Bennet, Dawson & Bartholomew, 1975*; *Cerreta et al., 2020*; *Molinaro et al., 2022*). Our study shows that lactate concentration in *Cyclura carinata* seems to reach a plateau approximately 15 min after capture. If this weren't the case, we would have expected lactate values to be significantly higher in individuals captured with a noose even at $t_{15}$, something that is not apparent in our data. While we cannot be certain that lactate concentration wouldn't increase further with additional time in captivity, our data suggest that the effect of different capture techniques seems to dissipate after approximately 15 min in captivity. Yet, with this study we suggest that having the possibility to choose between two alternative capture techniques, one less stressful than the other, may still be of value when in need of sampling individuals with a

known health condition or when sampled individuals are only needed for a very brief amount of time (for example, only to get a positive ID on the individual by reading the PIT tag number). Such rationale finds its support in the potential outcome of an increased concentration of lactate in the blood. Individuals with high blood lactate values are physically exhausted (*Bennet, Dawson & Bartholomew, 1975*; *Bartholomew, Bennet & Dawson, 1976*; *Prezant & Jarchow, 1997*; *Divers & Stahl, 2019*) and could momentarily be unable to properly respond to environmental stimuli (*e.g.*, evading an approaching predator or conspecific competitor). Indeed, in healthy and robust individuals intense lactate acidosis symptoms will eventually fade, and lactate concentration will return to its physiological state and the individual to its full behavioral capacity. However, the recovery rate from the activity-induced metabolic acidosis is usually environmentally dependent (*Bennet, Dawson & Bartholomew, 1975*; *Gleeson & Bennet, 1982*; *Gleeson, 1991*; *Guppy et al., 1987*; *Hartzler et al., 2006*) and, in suboptimal conditions, individuals may take longer to return to their status prior to capture, potentially exposing them to unwarranted risks. Therefore, especially in situations where handling will be brief, the presence of introduced predators may represent a risk, or during times of intense intraspecific interactions, sampling without the use of a noose pole, when possible, should be preferred.

The Turks and Caicos rock iguanas sampled in this study were clinically healthy and robust. Most of the blood values were similar to those reported for other members of the Iguanidae family. For example, the average packed cell volume (PCV) reported for green iguanas, *Iguana iguana*, (*Harr et al., 2001*) was 36.7%, somewhat higher than the average found in our study for Turks and Caicos rock iguanas (29.9%). PCV for Turks and Caicos rock iguanas were closer to values reported for the basilisk lizard (*Basiliscus plumifrons*, 31.4%; *Dallwig et al., 2011*), the Galápagos common land iguana (*Conolophus subcristatus*, 28.35% *Lewbart et al., 2019*), and the Galápagos pink land iguanas (*Conolophus marthae*, 31.53%, *Colosimo et al., 2022*). Packed cell volumes less than 20% would indicate anemia (*Dallwig et al., 2011*), but the lowest value recorded in our study was 22%, indicating that none of the sampled iguanas seemed to suffer from this ailment. Blood sodium levels reported for green iguanas and basilisk lizards were 160 and 153.5 mmol/L respectively (*Harr et al., 2001*; *Dallwig et al., 2011*), very similar to the average sodium levels we found for *Cyclura carinata* (158 mmol/L). Another health status determining value is blood glucose. Basilisk lizards (that were held in cloth bags and sampled 12 h post-capture) had a moderately high mean blood glucose level of 203 mg/dL (*Dallwig et al., 2011*). Large terrestrial Galapagos iguanids had mean blood glucose levels (*Conolophus pallidus*, 135 mg/dL; *Conolophus subcristatus*, 126 mg/dL; *Conolophus marthae* 155 mg/dL; *Lewbart et al., 2019*; *Colosimo et al., 2022*) similar to that recorded for *Cyclura carinata* (150 mg/dL), whereas, Allen Cays rock iguanas (*Cyclura cychlura inornata*) had a mean blood glucose level of 189 mg/dL (*James, Iverson & Raphael, 2006*), somewhat higher than their Turks and Caicos congeners. While the i-STAT and other point-of-care analyzers require cautious interpretation due to their reportedly lower accuracy with non-mammalian species (*Ng et al., 2014*), the comparative analysis between *C. carinata* and other iguanids suggests consistency and reliability in the collected data and is consistent with a clinical picture of healthy individuals. A similar conclusion can be drawn

from the analysis of the SMI body condition index calculated for the sampled individuals. Only one iguana showed indications of suboptimal health: a female that was blind in the left eye and noticeably thinner than other iguanas sampled (Fig. S1; individual tagged as 656LW). While iguanas can rely on their sense of taste and smell to pinpoint and identify potential trophic resources, partial blindness represents an obvious hindrance in species that heavily rely on visual cues to locate food and could be a detriment in other aspects of daily life as well. Interestingly, only body condition index identified this female as an outlier, as every other parameter measured was well within the range observed for other individuals (Fig. 3).

While our sample size may not be ample enough to establish reference intervals for the parameters included in this study (Geffre et al., 2009), the results of this study present a useful resource for biologists, veterinarians, and other researchers studying and making efforts to conserve this and other related species of Caribbean iguanas. In particular, we showed that capturing iguanas using noose poles results in significantly higher levels of lactate right after the capture event. Our results suggest that, when possible, grabbing individuals by hand can spare them from a struggle that would significantly increase the lactate levels in their blood.

*Cyclura carinata* is an integral part of the Turks and Caicos Islands terrestrial ecosystem, and their viability is vital to the persistence of other species, both plant and animal. Reptile medicine poses challenges as reptiles conceal disease and irregularities, and a knowledge gap exists in terms of reference data. By determining baseline values for *C. carinata*, the approach to management and conservation of the species can be made more efficient. Awareness of the physiological impact of capture and handling can be used to determine the most appropriate sample collection time and technique. Similarly, the information provided here may aide treatment of other reptiles and the comparison of species standards. Because temperature plays an integral role in lactate metabolism, this environmental variable should be included in future studies to investigate how fast sampled individuals can re-establish their physiological condition after a population study.

## ACKNOWLEDGEMENTS

We thank the TCI Department of Environment and Coastal Resources, the Turks and Caicos National Trust, and Big Ambergris Cay for logistical assistance with field work. We also would like to express our gratitude to Kent Passingham for his logistical support. Finally, we would like to express our gratitude to Dr. Hayes and Dr. Faria for their insightful and thorough revisions of the manuscript.

### Funding

Funding for this study was provided by a Darwin Plus grant (DPLUS121) from the U.K. Department for Environment, Food & Rural Affairs to the Royal Society for the Protection of Birds, and by the San Diego Zoo Wildlife Alliance. The funders had no role in study design, data collection and analysis, decision to publish, or preparation of the manuscript.

## Grant Disclosures

The following grant information was disclosed by the authors:

Darwin Plus grant: DPLUS121.

Department for Environment, Food & Rural Affairs to the Royal Society for the Protection of Birds, and by the San Diego Zoo Wildlife Alliance.

## Competing Interests

The authors declare that they have no competing interests.

## Author Contributions

- Giuliano Colosimo conceived and designed the experiments, performed the experiments, analyzed the data, prepared figures and/or tables, authored or reviewed drafts of the article, and approved the final draft.
- Gwyneth Montemuro analyzed the data, authored or reviewed drafts of the article, and approved the final draft.
- Gregory A Lewbart conceived and designed the experiments, performed the experiments, authored or reviewed drafts of the article, and approved the final draft.
- Gabriele Gentile conceived and designed the experiments, performed the experiments, analyzed the data, authored or reviewed drafts of the article, and approved the final draft.
- Glenn Gerber conceived and designed the experiments, performed the experiments, authored or reviewed drafts of the article, and approved the final draft.

## Animal Ethics

The following information was supplied relating to ethical approvals (*i.e.*, approving body and any reference numbers):

San Diego Zoo Wildlife Alliance Institutional Animal Care and Use Committee (IACUC Proposal #21-011).

## Field Study Permissions

The following information was supplied relating to field study approvals (*i.e.*, approving body and any reference numbers):

Turks and Caicos Department of Environment and Coastal Resources.

## Data Availability

The data collected in the field and the blood chemistry value registered for all animals analysed is available in the Supplemental File.

## Supplemental Information

Supplemental information for this article can be found online at http://dx.doi.org/10.7717/peerj.17171#supplemental-information.

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
