# Peer review of "Hand grab or noose pole? Evaluating the least stressful practice for capture of endangered Turks and Caicos Rock Iguanas Cyclura carinata"

_PeerJ, doi:10.7717/peerj.17171_

## Round 0.1 · original submission · Major Revisions

Dear Dr. Colosimo,

Your manuscript titled "Hand grab or noose pole? Health assessment and evaluation of best practices for capture of endangered Turks and Caicos Rock Iguanas Cyclura carinata" was considered by two expert reviewers and based on their opinions and my review, the decision is “Major revisions”.

Please carefully read the reviewers’ comments and address them fully in your revised manuscript. Please also ensure that all review, editorial, and staff comments are addressed in a response letter and any edits or clarifications mentioned in the letter are also inserted into the revised manuscript where appropriate.

Please note that submitting a revision of your manuscript does not guarantee eventual acceptance, and that your revision may be subject to re-review by the reviewer(s) before a decision is rendered.

·

Basic reporting

I have some regret about agreeing to review this very interesting paper because the last thing I want to do is discourage the authors from publishing their interesting study. Unfortunately, the manuscript fails to provide a thorough analysis and presentation of the data, but I'm recommending a thorough revision rather than rejection. I want to strongly encourage the authors to reassess the data, improve the manuscript, and resubmit it. Here are some suggestions:

1. Given the amount of data obtained, the authors could certainly broaden the hypotheses tested and use more appropriate statistics, for which they should probably work with a statistician. Here are some examples:

a. Body condition - I'm not sure why the index used would be advantageous to simple OLS regression residuals (some explanation would be helpful), but I was surprised to see a graph illustrating sex differences without addressing the obvious sexual dimorphism in the main text (other than a brief remark in the Methods). Yes, I realize the point of the graph was to illustrate an outlier (which usually merits perhaps a sentence, not a graph), but I found the sexual dimorphism much more interesting. I suggest the authors use analysis of covariance (ANCOVA) to examine how sex, season, and body size might influence body condition.

b. Lactate production - I suspect the authors reached appropriate conclusions regarding the effect of capture method on lactate levels, but they should likewise rely on one or more multivariate models with additional variables rather than the two univariate tests (one for each sex) to properly control for other sources of variation that likely exist. A linear mixed model (LMM) or a repeated-meaures ANCOVA would be appropriate because these models accommodate a repeated measure (each animal was tested twice for lactate level). The independent variables and covariates could include time (t0, t15), sex, treatment (noose, hand), body size, and even heart rate and/or body condition (the two covaried, so could just pick one), all of which could potentially influence lactate production. Why not see which predictors are most important, especially since they have the data?

c. Heart rate - The authors found that heart rate and body temperature were associated (as they should be), but heart rate is an alternative (and probably more sensitive though less consequential) measure of acute stress and could be treated much the same as lactate production. Would it also reveal an elevated stress response to noosing?

d. Other blood parameters could be similarly assessed with multivariate models, though the manuscript could become unwieldy and lose its primary focus if too much space is devoted to describing the results. If the central focus is sex comparisons, as in Fig. 4 (which looks concise and informative, though I didn't expect anything like it after reading the Introduction), then estimated marginal means (EMMs) from analysis of covariance (ANCOVA) models with their associated 95% confidence intervals could be plotted similarly. EMMs control for other potentially influential variables like body size.

Using a multivariate approach, a single omnibus model could suffice to examine each of the dependent measures above, but multiple models could also be tested with different combinations of predictors and interactions, and these could be assessed with informatic theory (e.g., Akaike Information Criterion) to better determine which predictors (i.e. which model) best account for variation in the dependent measures. Something to consider.

2. More thought should be given to explaining what the objectives of the study were. In its current form, the authors indicate in the Introduction that they conducted a general health assessment and compared lactate levels for the two modes of capture, but results for body condition, temperature, and heart rate came as a surprise to the reader much later in the manuscript. Hypotheses related to the additional measures should be elucidated more clearly in the Objectives portion of the Introduction (last paragraph or two).

3. The authors should better support the importance and relevance of their main hypothesis regarding differences in lactate concentration. My unanswered questions: Why are lactate levels important? Do we have any reason to believe that elevated levels of lactate cause any harm at all? Does capture method even matter if the iguanas will be handled more than a few minutes, after which lactate methods may become similar regardless of capture method? Are there circumstances for which post-capture handling would be brief enough that capture method would make a difference? Are capture and handling any more detrimental (in terms of lactate generation) than other routine activities, such as intraspecific chasing and fighting? And lastly, are there circumstances in which capturing iguanas by hand or noose may be impractical? From personal experience, I've found that naive iguanas are far easier to capture by hand than those captured previously.

4. The authors need to broaden their use of the literature. There are a ton of studies on lactate concentrations associated with various activities and capture/handling in a wide range of saurians (lizards and snakes). The authors cited a single study of sea turtles and framed the expectation of high lactate levels 15 min after capture "based on previous literature on other reptiles" without citing any more papers, which really surprised me. Moreover, there are several published papers providing hematology and biochemistry values from various Cyclura (and other iguana) species that the authors neglected to cite. And if body condition is given more attention - why not, the authors have a lovely graph showing sexual differences - there are papers that address seasonal variation in the body condition of Cyclura, with precipitation and its effects on vegetation being important.

5. The figures could use more attention, as the font for the x,y labels and numbers are too small and the black lines and squiggles within the boxes of Fig. 3 distract while offering no benefit in clarity.

6. The Ethical Statement strikes me as being over the top, with some details having no bearing on the methods or results. Be mindful of using undefined abbreviations (like DVM - some readers won't know it's an academic degree). Here, you also expressed a misunderstanding of a fundamental goal in experiments: "limit[ing] our sampling to the minimum number of individuals that would let us reach a statistical significance." Many experimental biologists fail to recognize that our goal should be to find meaningful effects (assessed by effect sizes) rather than significant effects (assessed by p-values). With a sufficient sample size, even a trivially small effect can be significant, which is meaningless. Sample sizes and p-values are conflated, which is a huge and underappreciated problem with null hypothesis testing. To the extent possible, we really should be reporting and emphasizing effect sizes more so than p-values.

7. Bonferroni adjustments of alpha for multiple tests are on their way out, as they are overly conservative (inflating type II errors). A better approach would be to adjust p-values using false discovery rate (Benjamini and Hochberg, 1995). Several online calculators make doing so a breeze:

https://www.sdmproject.com/utilities/?show=FDR [EASIER]
https://tools.carbocation.com/FDR [MORE DETAILED]

8. Avoid unnecessary details like "When cross-referencing the ID of the individual identified by this analysis with field notes recorded while collecting samples..." Just telling us that the female in suboptimal condition had a blind eye tells us everything we need to know.

In sum, the authors have a great data set and can substantially improve the quality of their study. I would feel badly if my remarks discouraged them and led to abandonment of the manuscript. My goal is simply to encourage them to give the world a better paper - one that matches the quality of their study design and hard-won data.

Experimental design

See above

Validity of the findings

See above

·

Basic reporting

1. On line 75, the authors cite Mones et al. (2021) as “a study (…) comparing manual restraint techniques.” I did not manage to get access to the full text of this study, however, from the reading of the abstract, it seems it does not compare different manual restraint techniques, but the performance of two different analysers for lactate measurements. Please confirm this and correct/delete the statement if necessary.
2. On line 258 the authors use the term lactic acidosis instead of lactate acidosis, as previously used on lines 70 and 72. Should be changed for the latter for the sake of consistency. Even if lactic acid and lactate are sometimes used interchangeably they are not the same exact molecule (and by their usage of the terms, the authors seem to be aware of that difference, making this instance appear to have been a simple typographic error).

Experimental design

1. Establishing baselines for relevant physiological values is of definite importance, but the way the paper is initially presented makes it feel as centred around the “hand grab or noose pole” question, without really presenting a reason for that comparison. Noosing has been used a method of capturing lizards since beginning of the 20th century in the least (Eimer 1882; Dunn 1915), and as far as I know, no studies have shown it to be detrimental for the lizards in comparison to other methods. While I do not deny that comparing the methods could also be of importance, some reasoning or context as to why the authors decided to compare the methods could be included in the introduction.
—Duun, E.R. (1915). Notes on the habits of Sceloporus undulatus (Latreille). Copeia, 19, 9.
—Eimer, T. (1882). Mode of capture of lizards in southern Europe. Annals and Magazine of Natural History, 9(5), 138–140.

Validity of the findings

1. On line 245, the authors conclude “grabbing individuals by hand can spare them from a struggle that would significantly increase the lactate levels in their blood.” The choice of wording might induce the erroneous interpretation that using a noose pole will result in overall higher values of blood lactate; this is not what is shown by the author’s data. The highest values for blood lactate values are at t=15 with no difference between methods, thus what the author’s data shows is a significantly faster increase of the lactate levels in the lizards’ blood, although highest measured lactate accumulation levels are the same for both methods.
It may happen lactate blood levels do reach significantly higher overall values after noose pole capture, if values peak shortly after the struggle finishes, after t=0, but have already started lowering at t=15 (as shown by Gleeson & Bennet (1982) for Varanus salvator, after exhaustive treadmill running). However, the authors’ data is not enough to conclude such, thus reiterating the importance of more clear wording for their conclusion.
—Todd T. Gleeson, Albert F. Bennett (1982). Acid-Base Imbalance in Lizards During Activity and Recovery. Journal of Experimental Biology, 98 (1), 439–453.

---

## Round 0.2 · accepted · Accept

Dear Dr. Colosimo,

Thank you for submitting your revised manuscript titled " Hand grab or noose pole? Evaluating the least stressful practice for capture of endangered Turks and Caicos Rock Iguanas Cyclura carinata". After reading the revised manuscript and the reviewer’s comments (see attached) I’m happy to let you know that decision is “accept”.

There is one minor edit I would like you to include - Figure 3, supplementary figure 1 and L297 – please add to these figures legends and to the text how outlier(s) were determined.

·

Basic reporting

(no comment)

Experimental design

(no comment)

Validity of the findings

(no comment)

Additional comments

I would like to congratulate the authors on the very thorough work they put on improving the manuscript. The new version not only is much more pleasurable to read as I believe it conveys the objectives and findings of the study, as well as the effort put towards its development, in a much more clear way.

I have no further comments towards changing the paper and believe it meets the requirements to be accepted and published.